# Effect of Grazing Management on Predator Soil Mite Communities (Acari: Mesotigmata) in Some Subalpine Grasslands from the Făgăraş Mountains—Romania

**DOI:** 10.3390/insects14070626

**Published:** 2023-07-12

**Authors:** Minodora Manu, Raluca Ioana Băncilă, Marilena Onete

**Affiliations:** 1Department of Taxonomy, Ecology and Nature Conservation, Institute of Biology Bucharest, Romanian Academy, Street Splaiul Independenţei, 060031 Bucharest, Romania; 2Faculty of Natural and Agricultural Sciences, University Ovidius Constanţa, 900470 Constanţa, Romania; raluca.ioana.bancila@univ-ovidius.ro; 3Department of Biospeleology and Soil Edaphobiology, “Emil Racoviţă” Institute of Speleology, Romanian Academy, 050711 Bucharest, Romania

**Keywords:** environment, grassland, grazing, intensive, mite, soil

## Abstract

**Simple Summary:**

Grasslands are one of the most widely distributed ecosystems at a global scale. They provide a broad range of ecosystem services, such as provisioning, regulating, cultural and supporting. Regulating services are strongly connected with biodiversity. Grasslands have an important role in sustaining the biodiversity of Europe, being considered a model system for biodiversity analyses. One of the important components of biodiversity is soil fauna, mites especially. Grazing management has also had an impact on soil mite communities. In this context, the aim of this research was to demonstrate the influence of the management type and of environmental variables on the composition of mite communities. The study revealed that the highest species richness was found in ungrazed grasslands compared with intensively grazed ecosystems, where the overall number of these invertebrates was higher. The study demonstrated that air temperature and soil penetration resistance were significantly higher in intensively grazed grasslands, while air relative humidity and vegetation cover were significantly lower in intensively grazed grasslands. Considering the range of species, the significant differences between the two types of grassland management are due to the varied influences of environmental variables, especially of vegetation coverage and soil electrical conductivity. This study revealed that the management regime of grasslands influences the structure and the spatial dynamics of soil mites.

**Abstract:**

For the first time in Romania, a complex study was conducted on soil mite communities from two types of managed grasslands: ungrazed and intensively grazed. The study was accomplished in August 2018, in the Făgăraş Mountains. Within the soil mite communities (Mesostigmata), 30 species were identified, from 80 soil samples. The following population parameters were investigated: species richness, numerical abundance, dominance, Shannon index of diversity, evenness and equitability. Eight environmental variables were also measured: soil and air humidity; soil and air temperature; soil pH; resistance of soil to penetration; soil electrical conductivity; and vegetation coverage. The results revealed that species richness, Shannon index of diversity, evenness and equitability indices had higher values in ungrazed grasslands, whereas in intensively grazed areas, the numerical abundance and dominance index had significantly higher values. The species *Alliphis halleri* was dominant in the ungrazed grasslands. Each type of managed grassland was characterised by specific environmental conditions, which had an important influence, even at the species level.

## 1. Introduction

At the global scale, grasslands are among the most widely distributed terrestrial biomes [1]. Grasslands in Eastern Europe cover around 282,000 km^2^ and, of these, 70% are high natural value grasslands. Some natural or seminatural grasslands are found in European mountains above the timberline, where the growing season is too short to sustain forest growth [2]. The grasslands provide a broad range of ecosystem services, such as provisioning (food, medical plants, honey and pollen), regulating (climate regulation through carbon sequestration, protection against erosion and flooding, pollination of food crops), cultural (tourism, education, aesthetic values) and supporting (soil formation, nutrient cycling, photosynthesis) [1,3,4]. Regulating services are strongly connected with biodiversity. Grasslands have an important role in sustaining the biodiversity of Europe, through the presence of many rare and endangered plant and animal species [5]. Complex studies on these types of ecosystems have revealed that they are very suitable as a model system for biodiversity analyses due to: (a) their wide latitudinal and altitudinal distribution; (b) the small-scale species richness which varies from low to extremely high; and (c) the various drivers that influence diversity patterns, such as climate, spatial heterogeneity, the landscape configuration and the type of management [5]. The main threats to grassland biodiversity in Eastern Europe are changes in management, especially in mountainous areas [2]. Many European countries have already lost culturally and naturally important grasslands due to changes in management practices. According to Nita et al., 2019 [6], the intensive, highly specialised management of grasslands has led to a decrease in their biodiversity. Traditional management usually supports biodiversity and lower management intensity is associated with higher grassland multifunctionality [2,4].

In Romania, permanent grasslands cover 44,940 km^2^, and represent 18.90% of the land area of the country. More than half of this grassland area is degraded due to overgrazing [3]. According to Tőrők et al., 2020 [2], in Romania, subsidies have produced increased numbers of grazing livestock, which in turn has led to reduced grassland biodiversity. However, data are not available on the impact of nonoptimal livestock density [3,6].

One of the important components of biodiversity is soil fauna. The soil mesofauna (Acari, Collembola, Enchytraeidae, Nematoda) participate directly or indirectly in soil fertility and nutrient availability by mineralising organic matter. The soil fauna supports the growth of microbes and plants [7]. In recent decades, the demand for food has increased at a global scale. This has meant an intensification of agriculture, through intensive use of fertilisers and agrochemicals, as well as higher densities of livestock. The intensity of grazing by livestock also affects soil mesofauna and the soil processes that they drive, such as decomposition of soil organic matter, nutrient cycling and soil structure [8,9,10,11]. Mites in grasslands are one of the most abundant soil invertebrate groups, being valuable bioindicators for the terrestrial environment [8,12,13,14]. They belong to a broad range of trophic categories, i.e., parasites, herbivores, fungivores, detritivores, microbivores, scavengers or omnivores and predators [8,15]. Grazing management has also had an impact on soil mite communities, such as the predominantly predatory Mesostigmata. In Europe, studies have revealed that intense grazing affects these populations, decreasing their abundance and species richness [12,13,16,17,18,19,20].

Internationally visible research networks related to Romanian grasslands are still scarce [6]. Ecological studies on predatory soil mite communities from grassland ecosystems are few and fragmented. They have revealed the impact of heavy metal pollution or the influence of other environmental variables on these soil invertebrates in controlled experimental fields [14,21,22,23].

Taking into account all this information, the aims of the present research are to: (I) provide information on the impact of management type (intensive grazed vs. ungrazed grasslands) on the structure of soil mite communities; (II) quantify some environmental variables characteristic of grassland ecosystems; and (III) highlight the influence of the management type and environmental variables on the composition of mite communities. Therefore, the present study will bring new and original information about the ecology of predatory soil mite communities from two types of managed grassland, contributing to increasing the international visibility of research in Romania.

The main hypotheses of the present study are: (1) does the composition of the mite communities differ between the two types of grasslands, with a higher species richness of the invertebrate fauna and greater abundance in ungrazed ecosystems compared with intensively grazed plots, and (2) is each type of grassland characterized by bioindicator species?

## 2. Material and Methods

### 2.1. Study Area

The research was carried out in 2018, in the Făgăraş Mountains, Romania, which are part of the Meridional Carpathians. The Făgăraş Mountains have a total area of 2400 km^2^, being about 70 km east to west and 45 km north to south, and with a maximum altitude at Moldoveanu peak of 2544 m. They are heavily glaciated, with lakes, eroded peaks and morainic deposits [24]. The Făgăraș Mountains have the status of a protected area under the European ecological network Natura 2000 [25,26,27,28].

The present study was made in four alpine grasslands, with two types of management scenario (grazed by sheep or ungrazed): Sterminoasa, Cocorâciu, Vemeşoaia and Galbena (Figure 1).

The Sterminoasa ungrazed (Su) grassland is situated at 45°34′39.13″ N, 24°23′39.4″ E at 1752 m altitude, on a slope of 15° and with west exposure. The type of soil is lithosol dystric and regosol humbric-lithic. The vegetation was dominated by *Agrostis capillaris* L., *Deschampsia cespitosa* (L.) P. Beauv., *Festuca rubra* L. and *Poa media* (L.) Cav. [29] (Figure 2a).

The Sterminoasa intensively grazed (Sg) grassland is situated at 45°34′32.7″ N, 24°23′42.48″ E at 1751 m altitude, on a slope of 15°–20° and with west exposure. The type of soil is lithosol dystric and regosol humbric-lithic. The vegetation was dominated by *Agrostis capillaris*, *Deschampsia cespitosa*, *Festuca rubra*, *Nardus stricta* L., *Poa media* and *Trifolium pratense* L. [29] (Figure 2b).

The Cocorâciu ungrazed (Cun) grassland is situated at 45°34′42.97″ N, 24°22′19.53″ E at 1800 m altitude, on a slope of 5° and with south exposure. The soil type is regosol dystric-lithic. The vegetation was dominated by *Agrostis capillaris*, *Festuca rubra*, *Deschampsia cespitosa*, *Alchemilla glabra* Neygenf. (incl. *A. boleslai* Pawl.), *Nardus stricta*, *Plantago gentianoides* Sm., *Poa annua* L. and *Trifolium repens* L. [29] (Figure 2c).

The Cocorâciu intensively grazed (Cg) grassland is located at 45°58′50.4″ N, 24°41′16.9″ E at 1910 m altitude, on a slope of 5°–10° and with south exposure. The soil type is regosol dystric-lithic. The vegetation was dominated by *Agrostis capillaris*, *Deschampsia cespitosa*, *Nardus stricta*, *Poa annua* and *Trifolium repens* [29] (Figure 2d).

The Vemeşoaia ungrazed (Vun) grassland is located at 45°32′47.4″ N, 24°26′28.8″ E at 1777 m altitude, on a slope of 20° and with south exposure. The soil type is regosol dystric-lithic. The vegetation was dominated by *Agrostis capillaris*, *Deschampsia cespitosa* and *Nardus stricta* [29] (Figure 2e).

The Vemeşoaia intensively grazed (Vg) grassland is situated at 45°32′44.7″ N, 24°26′30.6″ E at 1747 m altitude, on a slope of 15° and with south exposure. The soil type is regosol dystric-lithic. The vegetation was dominated by *Nardus stricta*, *Festuca rubra* and *Vaccinium myrtillus* L. [29] (Figure 2f).

The Galbena ungrazed (Gun) grassland is located at 45°37′47.8″ N, 24°23′57.9″ E at 1710 m altitude, on a slope of 15°–20° and with west exposure. The soil type is lithosol dystric and regosol humbric-lithic. The vegetation was dominated by *Achillea distans* subsp. *stricta* Janch., *Nardus stricta*, *Festuca rubra* and *Veratrum album* L. [29] (Figure 2g).

The Galbena intensively grazed (Gg) grassland is located at 45°32′47.4″ N, 24°27′03.2″ E at 1750 m altitude, on a slope of 20° and with west exposure. The soil type is lithosol dystric and regosol humbric-lithic. The vegetation was dominated by *Agrostis capillaris, Deschampsia cespitosa* and *Nardus stricta* [29] (Figure 2h).

### 2.2. Mite Samples

Soil samples were collected with a MacFadyen soil core (with a diameter of 5 cm), at 10 cm depth. In total, 80 soil samples were collected (10 samples in each type of grassland), randomly and in undisturbed soil structure. The extraction of soil invertebrates took about 14 days. We used the Berlese–Tullgren extraction method [30]. Mite specimens were identified using a Zeiss stereomicroscope (Carl Zeiss Instruments SRL, Bucharest, Romania) and an Axioscope A1 Zeiss microscope (Carl Zeiss Instruments SRL, Bucharest, Romania) (for Mesostigmata mites). The specimens were preserved in ethyl alcohol (75–90%). For Mesostigmata mites, the taxonomic identification was made after clearing them in lactic acid. Taxonomic slides were used and some body parts were mounted in polyvinyl alcohol–lactic acid mixture (PVA) medium [15].

The mites of the Mesostigmata order were identified at the species level using published identification keys [31,32,33,34,35,36,37,38,39,40,41]. The identified species are in the mite collection of the Institute of Biology Bucharest, Research Station at Posada.

### 2.3. Environmental Variables

In order to correlate the mite fauna abundances with environmental variables at the time of fauna collecting, the following abiotic and biotic factors were quantified: vegetation coverage—VegCov (%); air temperature—Tair (°C); air relative humidity—Uair (%); soil temperature—Tsoil (°C); soil moisture content—H soil (%); soil electrical conductivity—CE (μS/cm); soil acidity—pH; and soil penetration resistance—RP (MPa). In order to measure all these variables, some field and lab equipment were used. In the lab, the pH and CE were measured with a C532 Jasco Consort pH-meter (Consort N.V., Turnhout, Belgium). The pH was measured as pH (H2O). Air temperature, air relative humidity, soil temperature and soil moisture content were measured in the field with a digital thermo-hygrometer PCE-310 (PCE Instruments UK Limited, Southampton, United Kingdom). Resistance to penetration was determined with a soil penetrometer, Step System GmbH, 41,010 (STEP Systems GmbH, Nuremberg, Germany), also in the field. The environmental parameters were measured at the same time as we collected the soil samples for mites. The number of measurements were similar to the number of collected soil samples (80 measurements for each environmental variable in all grasslands).

### 2.4. Data Analysis

All statistical analyses were performed with 4.0.3 R statistic software [42]. Plots were constructed using the package “ggplot2” [43]. To assess how well the sampling effort represented the true mite species community within the intensive grazed and ungrazed grasslands, we generated species accumulation curves. To compare diversity patterns, we first presented the data in the form of species abundance distribution curves. We then used analysis of variance (ANOVA) in the generalised linear mixed effects models (GLMMs) procedure to test the effect of grassland type and environmental variables on abundance, species richness and Shannon–Wiener diversity index (H′). We summed a Poisson error distribution with a log link function for abundance and species richness, and a Gaussian distribution with an identity link function for H′ index. Grasslands were included as a random effect to avoid pseudo-replication due to the paired sampling design. Original models included all environmental variables. We calculated variance inflation factors (VIFs) to check for problems with collinearity in predictors (corvif function) [44]. Tsoil and Uair showed VIFs > 3 and we removed them from the original models. Species accumulation and distribution curves were generated using the “BiodiversityR” package [45]. The GLMMs were fitted using glmer function of “lme4” package [46]. To avoid convergence warnings and singular models, we used the scale function from base R to centre and scaled the continuous variables [47]. To display the effects of environmental variables on mite species diversity graphically, based on predicted values from GLMMs, we used the “effects” R package [48].

We used betadisper to test whether intensively grazed and ungrazed sites were homogeneously dispersed in relation to their species in studied grasslands and adonis to test if the two grassland types had different species composition. In addition, we conducted nonmetric multidimensional scaling (NMDS) ordination with a Bray–Curtis similarity index to compare species composition between intensive grazed and ungrazed grasslands. We conducted a redundancy analysis (RDA) to investigate the effect of environmental variables on the composition of the mite community. All species composition analyses were conducted using the vegan package [49].

## 3. Results

In the investigated grassland ecosystems, 30 species of mites were recorded with 184 individuals.

We found that species accumulation curves did not approach an asymptote for either each intensively grazed or ungrazed grassland (Figure 3). This indicated that more samples are required to detect all mite species theoretically expected per each grassland type. Species data for intensively grazed and ungrazed grasslands are presented in the form of species abundance distributions in Figure 3. Considering the species *Alliphis halleri*, the higher values of its numerical abundance from intensively grazed ecosystems compared with ungrazed ones revealed that it could be considered an indicator species for the first type of grassland (Figure 4).

Abundance and species richness significantly increased with VegCov (Appendix A, Figure 5a,b). None of the other predictors significantly affected mite diversity, except CE, which had a positive significant effect on species richness (Appendix A, Figure 5c).

The betadisper test indicated that the dispersion of the intensively grazed and ungrazed grasslands was significantly different (*p* < 0.441) (Figure 6). NMDS showed that the species composition of intensively grazed and ungrazed grassland largely overlapped (Figure 7).

However, based on the adonis test, the β-diversity of Bray–Curtis distance between the intensively grazed and ungrazed grassland was significant (*p* < 0.027). The adjusted R2 of the RDA model was 0.09, which means that only 9% of the total variance of the mite community can be explained by the environmental variables in the intensively grazed and ungrazed grasslands. Among them, RP played a leading role in affecting the distribution of the mite community, followed by VegCov and CE (Figure 8).

Air temperature and soil penetration resistance were significantly higher in intensively grazed than in ungrazed grasslands, while air relative humidity and vegetation cover were significantly lower in grazed grasslands compared to ungrazed ones (Table 1, Figure 9). No significant difference was found between intensively grazed and ungrazed grasslands for the other environmental variables (Table 1).

## 4. Discussion

Comparing the number of species from the two types of grasslands (ungrazed vs. intensively grazed), we observed that 22 species were identified in the ungrazed grass but only 14 species in that intensively grazed. The same pattern was observed for the Shannon index of diversity. The highest value was recorded in the ungrazed grasslands (Appendix A). Upon analysing the number of individuals (species abundance), the situation changed (Figure 3). The grazed grasslands recorded the highest value of this parameter (115 individuals), compared with 69 individuals from ungrazed ecosystems. This difference is due to the numerical dominance of *Alliphis halleri* (64 individuals). This species is a sapro-coprophilous detriticole mite, showing wide ecological plasticity, distributed from lowlands up to the subalpine zone (1750 m in Západné Tatry Mountains, Slovak Republic, or at 2355 m in the Rheatian Alps, Italy). It has a wide European distribution, being found in France, Germany, Greece, Ireland, Latvia, the Netherlands, Poland, Slovakia, Spain, Sweden and Great Britain [50]. It is phoretic on many coprophilous insects, especially Scarabaeidae. It prefers substrates that contain high proportions of manure or decaying organic matter, such as meadow soils and soil detritus in pastures. It can colonise fresh dung with a high content of water and nitrates, as well as older dung at later stages of succession. On the other hand, *Alliphis halleri* can be found in soil detritus that does not contain measurable manure [39]. Species from the Eviphididae family, such as *Alliphis halleri*, belongs to “r”—selected organisms. These species are described as r-strategists, and live in unstable and disturbed environments. They have a high fecundity rate and ability to reproduce rapidly (exponentially). This reproduction strategy determines an increased number of individuals, but they are not resistant to predation or to the rough conditions of a disturbed ecosystem. The numerical dominance of this species in grazed grasslands was highlighted by the highest value of the dominance index (0.348) and by the lowest values of the evenness (0.341) and equitability indices (0.593), in comparison with ungrazed ecosystems (0.120, 0.604 and 0.837, respectively) (Appendix A). 

Different ecological studies have been conducted in many European countries focussing on soil invertebrate fauna, including mites, in grasslands under different management regimes. These studies include: (a) the effect of grazing by geese, goats and fallow deer has been studied on soil mites or the seasonal dynamics of mites in pastures and meadows in Poland; (b) the mite (Acari: Oribatida, Mesostigmata) assemblages associated with Lasius flavus (Hymenoptera: Formicidae) nests and surrounding soil in an Irish grassland; (c) research on macrochelid mites (Acari: Mesostigmata) occurring in animal droppings in a pasture ecosystem in central Italy; (d) the effect of climate changes on soil fauna communities from extensively used pastures in Germany; and (e) an evaluation of mesofauna communities as soil quality indicators in a national-level monitoring programme in the United Kingdom [7,20,51,52,53,54]. All studies revealed that the structures and dynamics of mite communities differ in accordance with the type of grassland management. The number of species obtained by us in ungrazed grasslands from the Făgăraş Mountains, Romania, is comparable with that obtained in upland seminatural or improved conventional ecosystems in Ireland (where between 20 and 27 species of mesostigmatid mites were identified) or with that from inland meadows in Latvia [13,55]. The results from Romanian intensively grazed grasslands are comparable with those from Germany or from semi-improved upland grasslands in the United Kingdom, where 13–15 mite species were identified [56,57]. In Romania, other similar studies revealed the presence of 23 Mesostigmata species in pastures from the Central Moldavian Plateau, 28 species in overgrazed grassland ecosystems from the Transylvania region or from 5 species in natural unfertilised grazed grasslands to 30 species in fertilised grazed ecosystems [14,21,58,59].

Considering the taxonomic spectrum from both types of grasslands in the Făgăraş Mountains, we observed that 43.33% of identified mites were characteristic of ungrazed ecosystems (Appendix A). The majority of them were free-living, very mobile predators, always searching for food (e.g., *Pergamasus crassipes*, *Pergamasus quisquiliarum*, *Vulgarogamasus kraepelini*, *Geolaelaps aculeifer*, *Geolaelaps nolli*). They prefer soils rich in organic matter with large pores [56]. *Geolaelaps aculeifer* and *G. nolli* are species tolerating dry ecosystems, being identified in grasslands with high values of vegetation cover [14]. In ungrazed ecosystems, only 26.66% of identified species were characteristic of these grasslands (Appendix A). Common species for all grasslands were *Arctoseius cetratus*, *Asca bicornis*, *Geolaelaps praesternalis*, *Veigaia nemorensis* and *Zercon carpathicus*. *Artoseius cetratus* prefers sandy soils and is characterised by a high reproduction rate, short development time and tolerance to chemical contamination of soil, being found in fertilised or heavily polluted grasslands [14,22,23,56,60,61]. The same ecological preferences were identified for species *Asca bicornis* and *Geolaelaps praesternalis*, common mites found in different types of grasslands (natural, fertilised or heavy metal polluted). They have been recognised as colonising species found in the first stage of succession in derelict industrial lands [60,61]. On the other hand, according to Madej, 2008, *Arctoseius cetratus* and *Alliphis halleri* dominated young pastures at the beginning of ecological succession, being r-selected and multivoltine [62].

Analysing the environmental parameters, we observed that the ungrazed grasslands were characterised by the highest average values of air temperature, vegetation coverage, soil resistance at penetration and soil electrical conductivity, and a lower value of soil pH. A high percent of vegetation coverage is a characteristic feature for ungrazed ecosystems and is very often directly correlated with an increased value of soil resistance at penetration, possibly due to a well-developed root system. The vegetation cover affects the horizontal and vertical patterns of variability in the soil penetration resistance. At the same time, penetration resistance is a physical property of soil that depends on the soil moisture content, porosity and permeability, the mineral and organic matter content of soil, pH, and the cation-exchange capacity [63]. A high value of soil electrical conductivity means that water-soluble salts (an important indicator of mineral nutrients in the topsoil) are available for plants, allowing their biomass to develop.

The intensively grazed grasslands were characterised by the highest average values of air humidity, soil temperature and moisture content. Grazing can influence the soil microclimate (especially soil temperature and moisture content) through some mechanisms: by increasing the radiant energy reaching the soil leading to higher temperatures, and by reducing the transpiration area of the vegetation, which reduces the rate of soil moisture loss [64]. On the other hand, grasslands are structurally and functionally heterogeneous habitats. Grassland soil temperature and moisture vary spatially, due to its topography, and temporally, due to seasonal changes in air temperature and precipitation. Grazing and seasonal variation in precipitation and temperature are important controls of soil and plant processes in grasslands [12,14,21,65].

The different structures of soil mite communities from ungrazed and intensively grazed ecosystems are due to the varying influence of environmental variables. The main factors that significantly influenced the numerical abundance and species richness were vegetation coverage, soil resistance at penetration and soil electrical conductivity. The relationship between soil mite communities and vegetation coverage was highlighted in other studies, revealing that a rich herbaceous layer and higher plant diversity lead to a higher quantity of organic matter, which are favourite microhabitats and food reservoirs for predatory soil invertebrates, such as mites [14,66]. The prey of Mesostigmata originates from different trophic levels, such as: primary decomposers, secondary decomposers and other predators [67]. The trophic position of Mesostigmata species is not significantly correlated with body size [68]. Species from the Lealapidae family (such as *Geolaelaps aculeifer*, *G. nolli*), Ascidae (such as *Arctoseius cetratus*, *A. eremitus*, *Asca bicornis*) and Parasitidae (such as *Pergamasus crassipes* or *P. quisquiliarum*) prefer free-living nematodes as food, their diet having a positive effect on the reproductive parameters and developmental times of these species [69]. Species from the Pachylaelapidae family (such as *Onchodellus alpinus*) prefer prey such as Diptera larvae and Enchytraeidae [67,68]. The food preferences could influence their reproductive life. Species such as *Alliphis halleri*, *Asca bicornis*, *Arctoseius cetratus*, *Geolaelaps aculeifer* and *G. nolli* are grouped in r-selected colonizers (which means a maximum reproductive capacity); meanwhile, species from the Veigaiidae, Zerconiidae or Pachylaelapidae families (such as *Veigaia nemorensis*, *Zercon carpathicus*, *Onchodellus alpinus*) are classified as k-selected persisters [70]. Anthropogenic pressure (such as intensive grazing) will reduce the whole mite communities’ species diversity and the r-strategists will become dominant (such as the numerical dominance of *Alliphis halleri* in grazed grasslands). Intensive grazing is one of the main anthropogenic factors that influence the percentage vegetation coverage directly. Diversity and the abundance of mite invertebrates were positively correlated with above-ground plant biomass [9,10,59,71]. Risch et al., in 2007, declared that grazed grasslands are characterised by a spatio-temporal variability in decomposition, which is positively related to soil moisture content and soil C and N concentrations [65]. Intensively grazed grasslands in the Făgăraş Mountains were characterised by the highest values of soil moisture content and soil temperature, factors that accelerate the decomposition process, making available food resources for a higher numerical abundance of Mesostigmata mites.

The distribution of soil mite communities is influenced by the soil resistance at penetration, soil electrical conductivity and soil pH. The distribution of predatory mite communities in different soil layers depends directly on soil porosity, which is correlated with soil resistance at penetration. A higher value of this factor in ungrazed ecosystems (possibly due to a rich root system) will decrease the vertical mobility of these invertebrates or increase the number of species with a smaller body size, which can migrate through small pores in soil [63]. Soil mites have considerable passive dispersal abilities. Diffusion barriers, such as a lack of continuous inter-connection among soil pores, might affect the active movements of some soil mite species [72,73,74]. Species from ungrazed grasslands, such as *Arctoiseius eremitus*, *Asca bicornis*, *Geolaelaps nolli* and *Iphidonopsis sculptus* recorded smaller dimensions (body lengths between 300–520 µm) [31,33,60].

One of the most important soil properties that determines micronutrient availability and leach ability is pH. An acid soil increases the micronutrient availability for plants, and enhances mobility of Cu, Fe, Mn and Zn [75,76]. Soil electrical conductivity is another abiotic parameter that provides information regarding the presence of salts or mineral nutrients in soil. The presence of a rich soil mite fauna in ungrazed grasslands enhances nutrient mineralisation and plant nutrient uptake [19,71,72,77,78].

Considering the effect of environmental variables at the species level, we observed that soil pH influenced *Arctoseius cetratus*, air humidity influenced both *Veigaia exigua* and *Alliphis halleri,* and soil temperature affected *Leiodinychus orbicularis*. *Arcoseius cetratus* is a species resistant to compaction, found in sandier soils with decaying organic matter rather than those from agricultural, ruderal sites and pastures. It is recognised as a pioneer of the early stages of the ecological succession of degraded lands due to its phoretic abilities. It has even been found in grasslands with high levels of pollution and with acid soils [14,60,61,79]. *Veigaia exigua* is a eurytopic species with a wide distribution in Europe, including Romania, from ultra-lowlands up to montane zones. It has frequently been identified in grasslands with *Agrostis capillaris* and *Achillea millefolium*. It also prefers habitats with a more humid climate and soils with a high humus content [21,22,78]. *Alliphis halleri* is a cosmopolitan mite species, inhabiting various habitats all over the world. Being a coprophilous species, in intensively grazed grasslands, it prefers microhabitats such as sheep manure, which is wetter and richer in organic matter. According to Błoszyk et al., 2020, *Leiodinychus orbicularis* rarely occurs in habitats such as xerophilous grassland, meadows or mixed deciduous forests [80]. Known in all of Europe, this species is more frequent in the nests of birds or small mammals, and in bat boxes, and dwells in the dung of these mammals.

## 5. Conclusions

In order to investigate the influence of the type of management of grasslands and their characteristic environmental variables on soil mite communities, four intensively grazed and four ungrazed ecosystems were analysed. For the first time in Romania, an extensive and complex study was conducted, focusing on this topic. In total, we analysed 80 soil samples, revealing the presence of 30 species of predatory mites (Mesostigmata). The study revealed that the highest species richness was found in ungrazed grasslands compared with intensively grazed ecosystems, where the overall number of these invertebrates was higher. Eight environmental variables were measured. We demonstrated that air temperature and soil penetration resistance were significantly higher in intensively grazed grasslands, while air relative humidity and vegetation cover were significantly lower in intensively grazed grasslands. Considering the range of species, the significant differences between the two types of grassland management are due to the varied influences of environmental variables, especially vegetation coverage and soil electrical conductivity. These influences were evident even at the species level. Species such as *Arctoseius cetratus*, *Veigaia exigua*, *Alliphis halleri* and *Leiodinychus orbicularis* are influenced by environmental parameters.

Considering all this information, we consider that the proposed aims of this study were demonstrated, revealing that the management regime of grasslands influences the structure and the spatial dynamics of soil mites. These soil invertebrates could be used, with success, as a biological instrument for the characterisation of grassland ecological status.

## Figures and Tables

**Figure 1 insects-14-00626-f001:**
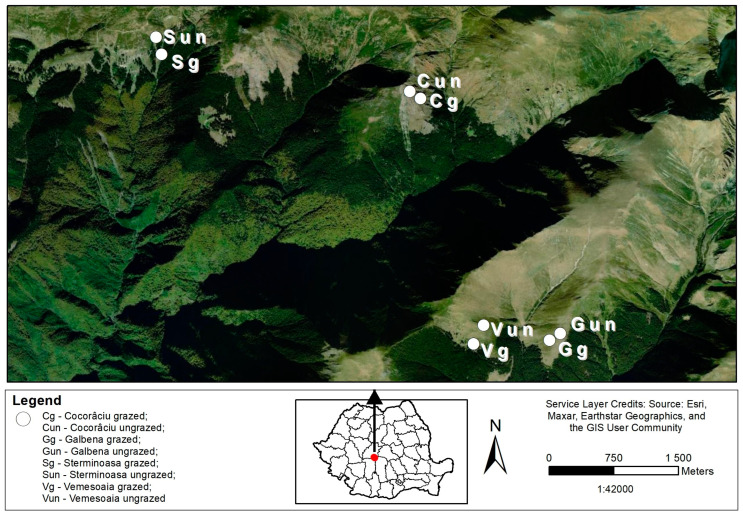
Geographical locations of investigated grasslands in the Făgăraş Mountains of Romania, 2018. Map created using ArcGIS software by Esri. (ArcGIS and ArcMap are the intellectual property of Esri and are used herein under licence. Version number: 10.4.0554. Copyright Esri. All rights reserved. For more information about Esri software, please visit www.esri.com (accessed on 27 March 2023). Base-map service layer credits: Esri, HERE, DeLorme, Intermap, increment P Corp., GEBCO, USGS, FAO, NPS, NRCAN, GeoBase, IGN, Kadaster NL, Ordnance Survey, Esri Japan, METI, Esri China, swisstopo, MapmyIndia, OpenStreetMap contributors, and the GIS).

**Figure 2 insects-14-00626-f002:**
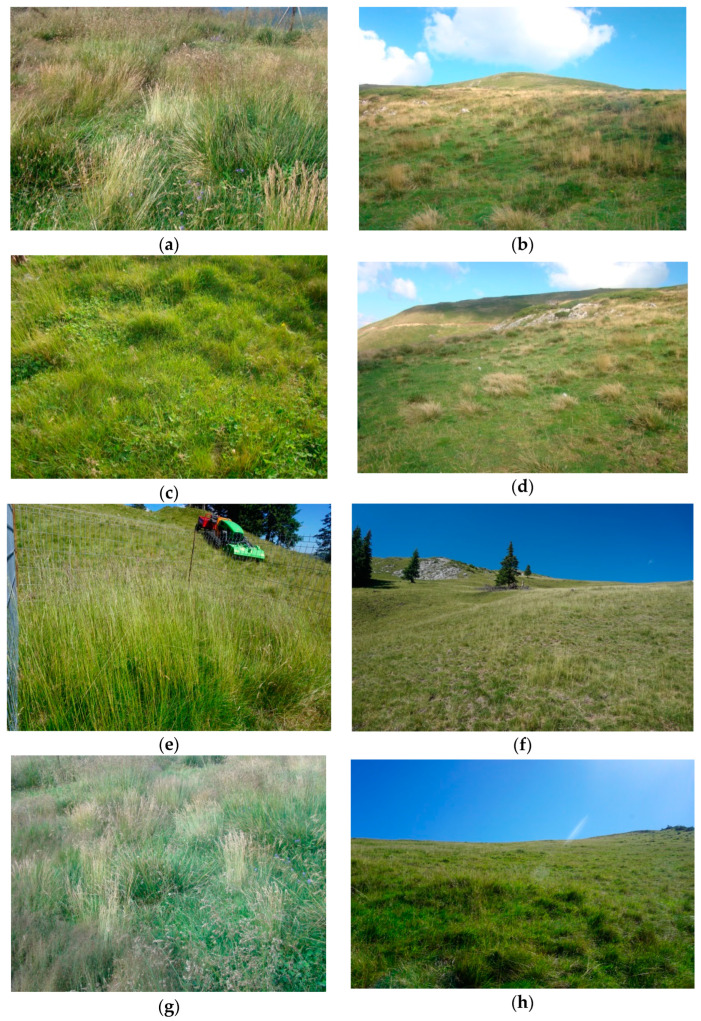
The general aspects of the investigated grassland ecosystems from Făgăraş Mountains—Romania, 2018 ((**a**) Sterminoasa intensively grazed; (**b**) Sterminoasa ungrazed; (**c**) Cocorâciu ungrazed; (**d**) Cocorâciu intensively grazed; (**e**) Vemeşoaia ungrazed; (**f**) Vemeşoaia intensively grazed; (**g**) Galbena ungrazed; (**h**) Galbena intensively grazed).

**Figure 3 insects-14-00626-f003:**
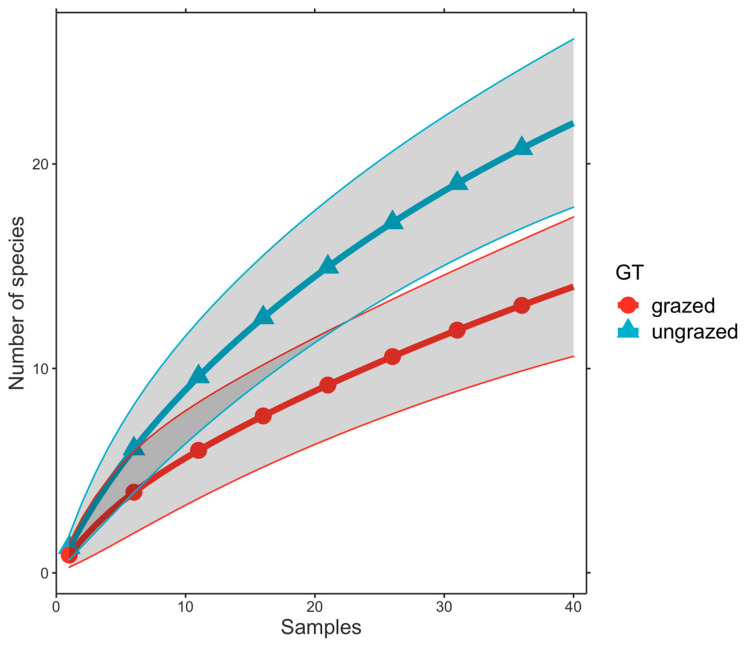
Mao Tau species accumulation curves for the two different grassland types (GT), i.e., intensively grazed and ungrazed.

**Figure 4 insects-14-00626-f004:**
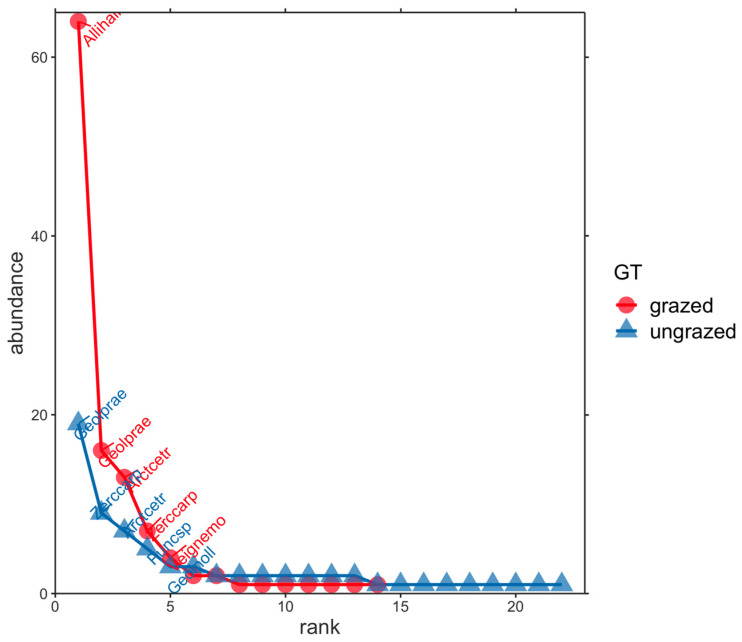
Species abundance distributions for the two different grassland types (GT), i.e., intensively grazed and ungrazed.

**Figure 5 insects-14-00626-f005:**
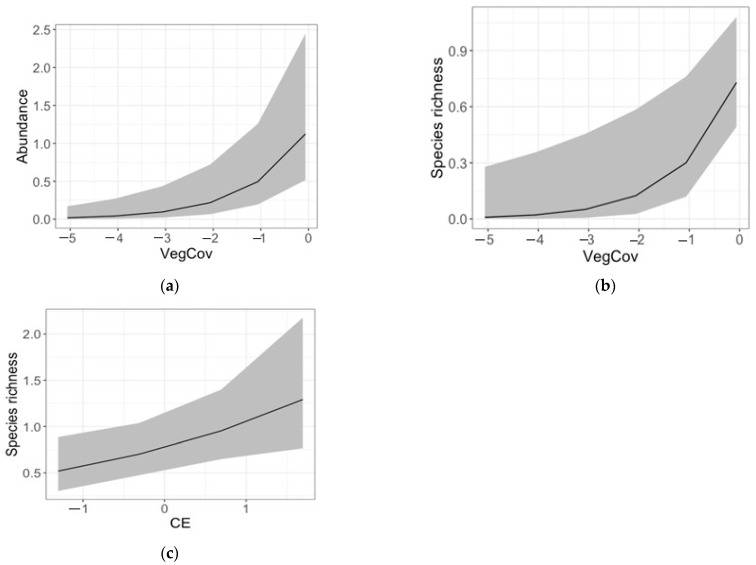
Effect of vegetation cover on abundance (**a**) and species richness (**b**) and the influence of soil electric conductivity on species richness (**c**) based on values predicted from generalised linear mixed models with a Poisson error distribution of residuals and log link function.

**Figure 6 insects-14-00626-f006:**
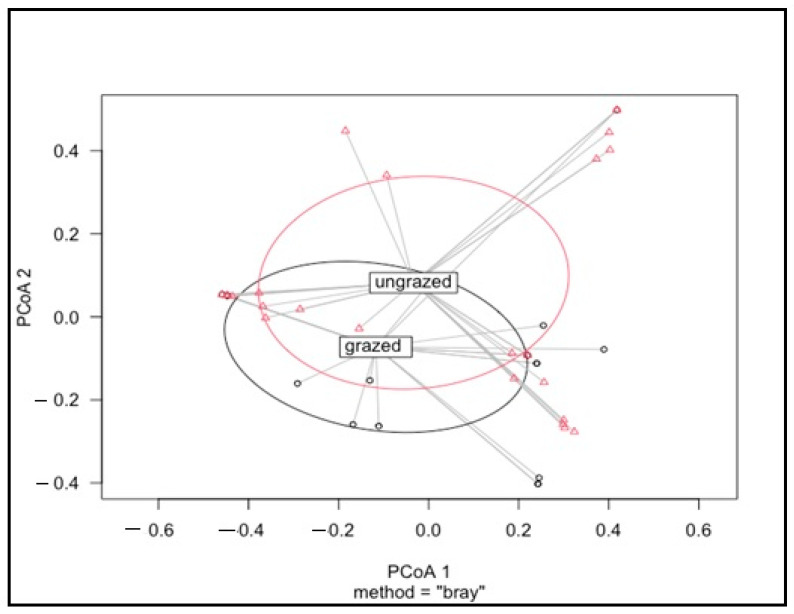
Mite community structure assessed by β-diversity patterns using the principal coordinate analysis plots of Bray–Curtis distances. Different colour and shape represent grassland types: intensively grazed (black circles) and ungrazed (red triangles). Adonis analysis was used to test the significance between groups (i.e., grazed and ungrazed).

**Figure 7 insects-14-00626-f007:**
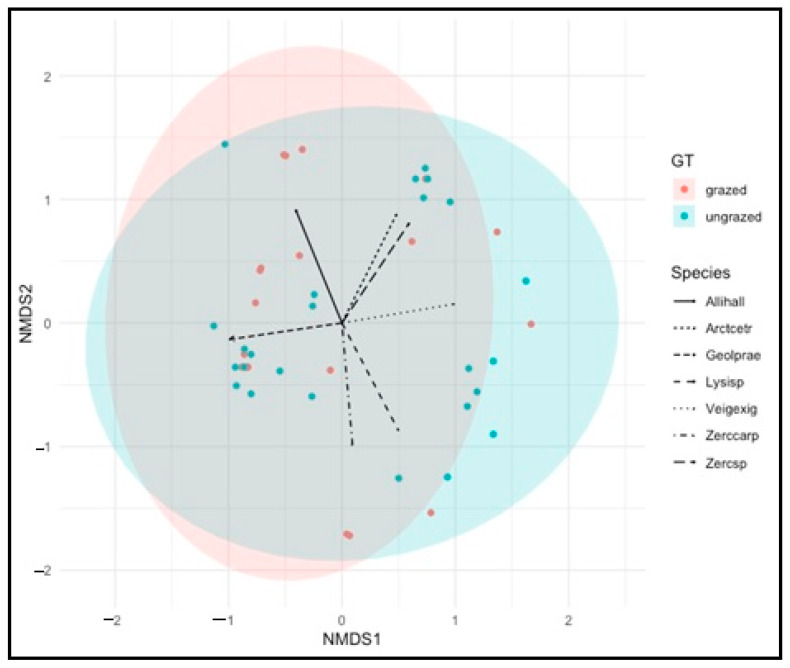
Nonmetric multidimensional scaling (NMDS) ordination (Bray–Curtis) of the mite species composition. Ellipses indicate the 95% confidence interval of intensively grazed and ungrazed grasslands. Species variables were fitted to NMDS axes and are plotted with arrows. Stress value was 0.287.

**Figure 8 insects-14-00626-f008:**
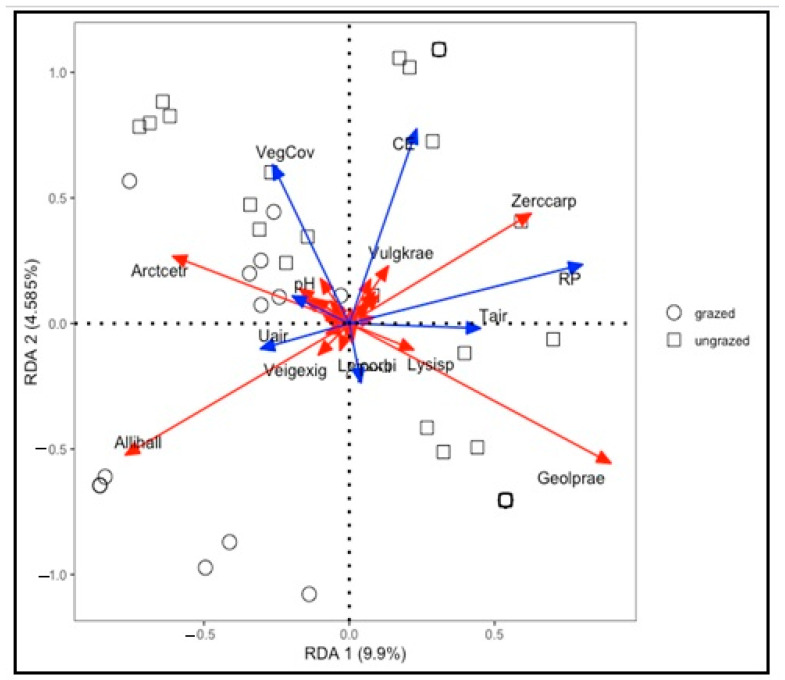
Redundancy analysis of the environmental variables on the composition of the mite community. Blue arrows refer to environmental variables, red solid lines refer to mite species, and open squares and circles refer to sampling points.

**Figure 9 insects-14-00626-f009:**
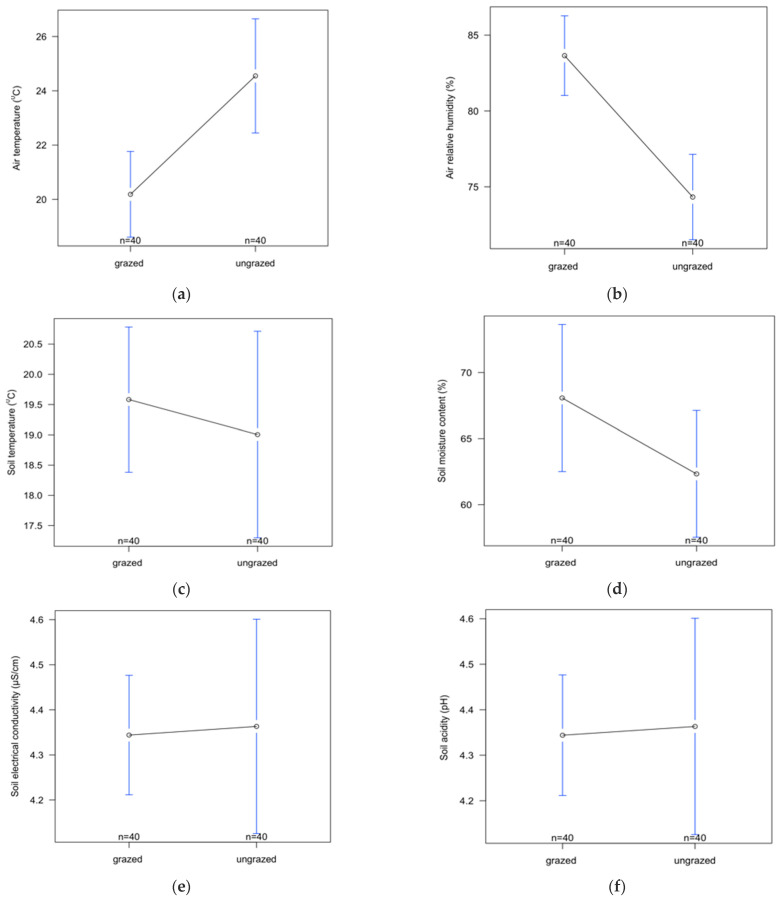
The variation in investigated environmental variables from intensively grazed and ungrazed grasslands ((**a**) air temperature, (**b**) air relative humidity, (**c**) soil temperature, (**d**) soil moisture content, (**e**) soil electrical conductivity, (**f**) soil acidity, (**g**) soil penetration resistance, (**h**) vegetation coverage). The open circles represent means and the blue lines represent the confidence intervals.

**Table 1 insects-14-00626-t001:** One-way Anova results of environmental variables. df = degrees of freedom; MS = mean squared; F = F statistic; *p* = level of significance.

Environmental Variables	Df	MS	F	*p*
Air temperature (°C)	1	380.63	11.267	0.001
Air relative humidity (%)	1	1739.11	24.018	<0.001
Soil temperature (°C)	1	6.6701	0.313	0.577
Soil moisture content (%)	1	659.53	2.497	0.118
Vegetation cover (%)	4	1087.81	16.518	<0.001
Soil acidity (pH)	1	0.007	0.021	0.887
Soil penetration resistance (Map)	1	2.71879	19.765	<0.001
Soil electrical conductivity (µS/cm)	1	150.70	0.199	0.657

## Data Availability

Data are contained within the article or Appendix A.

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
