# Peer review of "Effect of Grazing Management on Predator Soil Mite Communities (Acari: Mesotigmata) in Some Subalpine Grasslands from the Făgăraş Mountains—Romania"

_insects, 2023, doi:10.3390/insects14070626_

Round 1
Reviewer 1 Report
I am happy to see the elaboration of the manuscript. But, it has some scientific and grammatical issues in the current version, and I have highlighted some issues here in the attached MS file. Therefore, the present draft needs minor revision before further process.

Author Response
Comments: It has some scientific and grammatical issues in the current version, and I have highlighted some issues here in the attached MS file
Answer: All comments from the manuscript were followed. The proposed modifications (scientific and grammatical) from the text were checked! Please, see the manuscript!
Thank you very much for your time, effort and precious comments!
Reviewer 2 Report
In the future, 10 soil samples from one meadows seem to be not enough.
Author Response
Comment: In the future, 10 soil samples from one meadow seem to be not enough!
Answer: Thank you very much for the precious information! In the future I will consider that advice. Thank you very much for your time and effort!
Reviewer 3 Report
EFFECT OF GRAZING MANAGEMENT ON PREDATOR SOIL MITES COMMUNITIES (ACARI-MESOTIGMATA) IN SOME SUBALPINE GRASSLANDS FROM THE FĂGĂRAŞ MOUNTAINS – ROMANIA
The paper focusses on the effect of grazing on the soil mesostigmatic mite composition. However, at the end of the introduction three aims of the paper are listed, but I miss testable hypotheses: what do the authors expect to occur with (intensive) grazing, which component of the mesostigmatic mite fauna is thought to decrease or increase and can it also change the species composition?
First sentence under 2.2 is repetitive, delete this.
It is not clear how often the variable weather data are collected (air temperature, soil temperature, humidity, soil moisture) as these are extremely variable. If only measured at the date of sampling, these data are rather useless for predictions or analyses. If the measurement is continuous, please specify the period and how you ordered the data. Is pH measured as pH[H2O] or with salt addition (e.g. NaCl 0.1 mol, or KCl 1.0 mol)? How many samples are measured (same as those for the microarthropods?). How many measures are done with the penetrometer? Please specify.
I don’t understand the meaning of the sentence: ‘Focussing on the presence of Alliphis halleri, variation in species numbers between intensive grazed and ungrazed grasslands is evident (Figure 4).’ Why must I focus on one species to see differences? It is the most abundant one in grazed grasslands. Is the English correct here?
P. 8 on top: a % explained variance of only 9 is very low, so these environmental variables do not explain the species composition.
First at the Discussion, the number of species and identified individuals is mentioned (30 species over 184 individuals). These are primary results, so put it there. Nevertheless, the data are rather meager, especially the number of individuals is extremely low (on average a little more than two per sample!).
An option is to group the species to feeding guilds (nematode predators, collembolan predators, general predators) or also the life history strategies (number of generations, phoretics as adult or juvenile, etc.) to get some more information out of these data.
Last paragraph of p. 12, several common species are mentioned. Given the overall low numbers, how ‘common’ were these?
In short: grazed grasslands are trampled, thus have a higher soil compaction, hence another temperature and moisture regime. Not really breaking news.
Some sentences don't seem to contain the proper English words or meaning. Check on typo's.
Author Response
Comments: The paper focuses on the effect of grazing on the soil mesostigmatic mite composition. However, at the end of the introduction three aims of the paper are listed, but I miss testable hypotheses: what do the authors expect to occur with (intensive) grazing, which component of the mesostigmatic mite fauna is thought to decrease or increase and can it also change the species composition?
Answer: The main hypotheses of the present study are: does the composition of the mite communities differs between the two types of grasslands, with a higher species richness of the invertebrate fauna and more abundant in ungrazed ecosystems compared with intensively grazed plots and 2) is each type of grasslands characterized by bioindicator species?
I inserted in the manuscript the above paragraph.
Comment: First sentence under 2.2 is repetitive, delete this.
Answer: I delete it!
Comments: It is not clear how often the variable weather data are collected (air temperature, soil temperature, humidity, soil moisture) as these are extremely variable. If only measured at the date of sampling, these data are rather useless for predictions or analyses. If the measurement is continuous, please specify the period and how you ordered the data. Is pH measured as pH [H2O] or with salt addition (e.g. NaCl 0.1 mol, or KCl 1.0 mol)? How many samples are measured (same as those for the microarthropods?). How many measures are done with the penetrometer? Please specify. P. 8 on top: a % explained variance of only 9 is very low, so these environmental variables do not explain the species composition.
Answer: The environmental parameters were measured in the same time when we collect the soil samples for mites. The numbers of measurements are similar with the number of collected soil samples (80 measurements for each environmental variable in all grasslands). The penetrometer was used for each soil sample which was collected. The pH was measured as pH [H2O]. The variable weather data are measured in order to correlate the mite fauna abundances with environmental variables at the time of fauna collecting.
All these informations were inserted in the manuscript!
Comments: I don’t understand the meaning of the sentence: ‘Focusing on the presence of Alliphis halleri, variations in species numbers between intensive grazed and ungrazed grasslands is evident (Figure 4).’ Why must I focus on one species to see differences? It is the most abundant one in grazed grasslands. Is the English correct here?
Answer: This species was identified in both types of grasslands (intensively grazed and ungrazed). The higher values of its numerical abundance from intensively grazed ecosystems comparing with those ungrazed ones revealed that it could be consider as indicator species for the first type of grassland.
I changed the phrase as following: “Considering the species Alliphis halleri, the higher values of its numerical abundance from intensively grazed ecosystems comparing with those ungrazed ones, revealed that it could be consider as indicator species for the first type of grassland.”
Comment: First at the Discussion, the number of species and identified individuals is mentioned (30 species over 184 individuals). These are primary results, so put it there. Nevertheless, the data are rather meager; especially the number of individuals is extremely low (on average a little more than two per sample!).
Answer: In the results I put the sentence: “In the investigated grassland ecosystems, 30 species of mites were recorded with 184 individuals”.
I appreciate the reviewer’s opinion. The data are meager due to the climate conditions (the mite samples were collected in the summer and the soil was quite dry) on the one hand, and the grazing regime affects the soil and vegetation layers, on the other hand.
Comment: An option is to group the species to feeding guilds (nematode predators, collembolan predators, general predators) or also the life history strategies (number of generations, phoretics as adult or juvenile, etc.) to get some more information out of these data.
Answer:
The following paragraph was inserted at discussion:
The prey of Mesostigmata originates from different trophic levels, as: primary decomposers, secondary decomposers and other predators [67]. The trophic position of Mesostigmata species is not significantly correlated with body size [68]. Species from Lealapidae family (as Geolaelaps aculeifer, G. nolli), Ascidae (as Arctoseius cetratus, A. eremitus, Asca bicornis), Parasitidae (as Pergamasus crassipes or P. quisquiliarum) prefer free-living nematodes as food, their diet having a positive effect on reproductive parameters and developmental times of these species [69]. Species from Pachylaelapidae family (as Onchodellus alpinus) prefer prey as Diptera larvae and Enchytraeidae [68, 69]. The food preferences could influence the reproductive life. Species as Alliphis halleri, Asca bicornis, Arctoseius cetratus, Geolaelaps aculeifer, G. nolli are grouped in r- selected colonizers (which means a maximum reproductive capacity), meanwhile species from Veigaiidae, Zerconiidae or Pachylaelapidae families (as Veigaia nemorensis, Zercon carpathicus, Onchodellus alpinus) are classified as k-selected persisters [70]. Anthropogenic pressure (as intensive grazing) will reduce the whole mite communities’ species diversity and the r-strategists will become dominant (as numerical dominance of Alliphis halleri in grazed grasslands).
[67] Potapov, A.M.; Beaulieu, F.; Birkhofer, K.; et al. Feeding habits and multifunctional classification of soil-associated consumers from protists to vertebrates. Biol. Rev. 2022, 97, 1057-1117.
[68] Klarner, B.; Maraun, M.; Scheu, S. 2013. Trophic diversity and niche partitioning in a species rich predator guild – Natural variations in stable isotope ratios (13C/12C, 15N/14N) of mesostigmatid mites (Acari, Mesostigmata) from Central European beech forests. Soil Biol. Biochem. 2013, 57, 327–333.
[69] Rueda-Ramírez D., Palevsky E., Ruess L., 2022. Trophic links between soil predatory mites and nematodes as a key component of conservation biocontrol. Zoosymposia. 2022, 22, 61.
[70] Ruf, A. A maturity index for predatory soil mites (Mesostigmata: Gamasina) as an indicator of environmental impacts of pollution on forest soils. Appl.Soil Ecol. 1998, 9, 447–452.
Comment: Last paragraph of p. 12, several common species are mentioned. Given the overall low numbers, how ‘common’ were these?
Answer: The term of “common” refers to their presence in both types of grassland ecosystems.
Comments: Some sentences don't seem to contain the proper English words or meaning. Answer: The English spelling and grammar was checked by our collaborator Owen John Mountford, a native English speaker and research ecologist at UK Centre for Ecology & Hydrology.
Thank you very much for your precious comments, effort and for your time!
Reviewer 4 Report
The manuscript submitted for review is well thought out and very interesting. In addition, it is also well-written, so I have found no mistakes in methodology or in the interpretation of the results. Finally, I highly recommend this manuscript for publication.
Author Response
Comments: The manuscript submitted for review is well thought out and very interesting. In addition, it is also well-written, so I have found no mistakes in methodology or in the interpretation of the results. Finally, I highly recommend this manuscript for publication.
Answer: Thank you very much for your appreciation. Thank you very much for your time and effort!
Reviewer 5 Report
The authors examined the mite fauna from 80 core samples obtained from Romanian grasslands and provided a detailed analysis with environmental data. Although some minor corrections are needed, the paper as a whole is worthwhile.
There are problems throughout, such as "°" being represented by a superscript of "0" and some of " ″ " being represented by " ′ ".
Also, "meters" are written in a mixture of "m" and "metres".
Author Response
Comments: There are problems throughout, such as "°" being represented by a superscript of "0" and some of " ″ " being represented by " ′ ".
Also, "meters" are written in a mixture of "m" and "metres"
Answer: All requested modifications were made in the manuscript! Please, see pages 3, 4 and 12! Thank you very much for your time and effort!
Round 2
Reviewer 3 Report
Some parts of the ms have been improved (method description, adding hypotheses), however the major lack of sufficient data to support the conclusions is not improved. Adding life-history information helps in interpretation of the results, however, this could have been more integrated in the entire paper.
Still some typos in the ms in the additions (for instance p. 4 on top, insertion 2nd line: differ instead of differs), please check all.
Author Response
Reviewer: Some parts of the ms have been improved (method description, adding hypotheses), however the major lack of sufficient data to support the conclusions is not improved. Adding life-history information helps in interpretation of the results; however, this could have been more integrated in the entire paper.
Still some typos in the ms in the additions (for instance p. 4 on top, insertion 2nd line: differ instead of differs), please check all.
Answer:
On discussion another paragraph was inserted: “Species from Eviphididae family, as Alliphis halleri belongs to “r” – selected organisms. These species described as r-strategists, live in unstable and disturbed environments.They have high fecundity rate and ability to reproduce rapidly (exponentially). This reproduction strategy determines an increased number of individuals, but not resistant to predation or on rough conditions of disturbed ecosystem.
The typos modifications were made in the manuscript (differ instead of differs- page 4; “it could be considered” instead of “it could be consider” – page 7). I checked all manuscript!
Round 3
Reviewer 3 Report
Still a meagre paper due to the limited amount of animals caught, but publishable in its present form.